# Risk management in POCT blood glucose monitoring: FMEA approach aligned with ISO 15189:2022

Xiagang Luan[1ම], Lingling Ke[2ම], Minxuan Feng[3,4ම], Weiqun Peng[5], Houlong Luo[3], Hao Xue[3]*, Yong Xia[3]*

1 Department of Burns, Wuhan Third Hospital, Tongren Hospital of Wuhan University, Wuhan, China, 2 Department of Laboratory, Wuhan Third Hospital, Tongren Hospital of Wuhan University, Wuhan, China, 3 Department of Laboratory Medicine, Peking University Shenzhen Hospital, Shenzhen, China, 4 The First School of Clinical Medicine, Guangdong Medical University, Zhanjiang, China, 5 Department of Endocrinology, Peking University Shenzhen Hospital, Shenzhen, China

ම These authors contributed equally to this work.
* 184337543@qq.com (HX); xiayong@pkuszh.com (YX)

## Abstract

### Objective

Point-of-care testing (POCT) blood glucose meters provide rapid and convenient monitoring for clinical care and chronic disease management. However, their accuracy is often compromised by risks associated with personnel, equipment, and procedural inconsistencies. This study systematically assesses these risks using the Failure Mode and Effects Analysis (FMEA) method and proposes control measures aligned with ISO 15189:2022 standards.

### Methods

This study evaluated the risks associated with POCT blood glucose meters in clinical laboratory settings, encompassing the pre-analytical, analytical, and post-analytical phases. A multidisciplinary team employed FMEA to identify potential failure modes and their impacts. A risk matrix classified risks based on probability and severity, with "unacceptable" risks prompting targeted control measures. A follow-up assessment conducted three months later evaluated the effectiveness of these measures through feedback collection and quality control data analysis, ensuring effective risk mitigation in POCT practices.

### Results

The risk assessment identified distinct issues at each hospital: Peking University Shenzhen Hospital faced significant risks related to inadequate performance verification prior to hospital entry, insufficient personnel training, and data management problems, while Wuhan Third Hospital primarily encountered challenges with inadequate training and insufficient calibration and inadequate quality control. Control measures implemented at Peking University Shenzhen Hospital included stringent validation protocols, comprehensive training systems, and automated data management. At Wuhan Third Hospital,

**Data availability statement:** All relevant data are within the manuscript and its Supporting Information files.

**Funding:** The author(s) received no specific funding for this work.

the focus was on enhancing training oversight and establishing rigorous quality control measures and calibration Schedule. These interventions effectively reduced unacceptable risks and improved the safety and reliability of the monitoring process.

## Conclusion

Integrating FMEA with ISO 15189:2022 provides a structured approach for identifying and mitigating risks in the use of POCT blood glucose meters. Implementing tailored measures significantly enhances POCT accuracy and reliability, offering clinical institutions effective strategies to improve quality and ensure better patient outcomes.

## Introduction

Point-of-care testing (POCT) blood glucose meters are essential in clinical care and chronic disease management due to their rapid testing capabilities and portability, facilitating immediate decision-making at the patient's bedside [1]. This convenience enhances diagnostic efficiency, enabling healthcare providers to promptly adjust treatment plans and support patient self-management, ultimately improving health outcomes [2,3]. However, these devices pose risks, including operational errors such as improper sample handling, inadequate calibration, and poor data management, which can compromise test accuracy and patient safety [4–6]. Therefore, robust quality management practices tailored to POCT blood glucose meters are imperative.

ISO 15189:2022 addresses these challenges with guidelines focusing on critical aspects of POCT services, including organizational management, equipment selection, personnel training, and quality assurance [7]. These standards mandate comprehensive training programs to ensure personnel proficiency and emphasize routine maintenance and calibration to guarantee accuracy [8]. Clear service agreements are required to delineate roles and responsibilities among departments, ensuring consistent use in clinical laboratory settings.

To address the diverse risks associated with POCT blood glucose meters, a structured risk assessment framework is essential. Failure Mode and Effects Analysis (FMEA) is a proactive approach to identify, assess, and prioritize potential failure modes within medical devices and laboratory practices [9,10]. FMEA evaluates various factors—such as personnel competency, instrument functionality, reagent stability, and quality control practices—allowing for a comprehensive assessment of potential failure locations and effective mitigation strategies [9,11–14].

Recent research underscores the significance of quality management in POCT practices. For example, implementing the Plan-Do-Check-Act (PDCA) cycle enhances both the accuracy of blood glucose measurements and patient satisfaction [15,16]. This cyclical approach ensures continuous improvement through planning interventions, monitoring outcomes, and refining procedures based on feedback. Establishing quality indicators (QIs) is also crucial for laboratory quality assurance, providing measurable benchmarks to monitor risks and assess the testing process in alignment with ISO 15189:2022 standards [17].

Clinical nurses and laboratory personnel are vital in maintaining the accuracy of POCT blood glucose tests. Nurses execute quality control measures, adhere to standard operating procedures, and participate in regular training [18]. Laboratory professionals select appropriate instruments, perform calibration, and troubleshoot discrepancies between POCT and central laboratory findings [19]. This multidisciplinary collaboration enhances risk management, ensuring that POCT results are reliable and clinically actionable.

Technological advancements have significantly improved POCT blood glucose meters, including developments like enzyme-free electrochemical sensors that enhance sensitivity and specificity. Additionally, integrating POCT devices with electronic information systems reduces manual entry errors and facilitates real-time data management [20,21]. However, challenges remain, particularly regarding discrepancies between POCT results and traditional laboratory methods for diagnosing complex conditions such as type 2 diabetes [22]. Addressing these challenges through a structured approach is essential for optimizing POCT use in clinical practice.

In this context, the study applies FMEA methodology, in accordance with ISO 15189:2022, to conduct a comprehensive risk assessment of POCT blood glucose meter usage in clinical laboratory settings. By systematically identifying, categorizing, and addressing these risks, the study proposes targeted control measures aimed at enhancing the safety, reliability, and effectiveness of POCT, thereby providing valuable evidence-based guidelines for quality improvement in clinical diagnostics.

## Materials and methods

### Materials

This study was conducted in a clinical laboratory setting to systematically evaluate the operational processes of point-of-care testing (POCT) blood glucose meters for potential risks. The assessment covered key elements across the pre-analytical, analytical, and post-analytical phases: personnel, instruments, reagents, methods, environment, and quality control. The POCT blood glucose meters utilized were standard models commonly employed in clinical practice. All reagents, calibration solutions, and consumables adhered to the specifications outlined in ISO 15189:2022. Personnel involved were certified clinical laboratory technicians with relevant experience, trained to comply with standardized operating procedures.

### Methods

The study employed Failure Mode and Effects Analysis (FMEA) to assess the risks associated with the clinical use of POCT blood glucose meters. FMEA is a systematic, proactive tool that identifies potential failure modes, analyzes their impacts, and recommends corrective actions to minimize clinical risks. The risk assessment was conducted following ISO 15189:2022, emphasizing equipment calibration, personnel competence, and data integrity.

### Risk identification

A multidisciplinary POCT Quality Safety and Risk Assessment Team conducted the risk identification process. The team comprised experienced POCT technicians from two hospitals, a clinical physician from a burn-specific ICU, a POCT manager, and nursing staff. This diverse group ensured a thorough evaluation of risks across different clinical laboratory settings.The team employed brainstorming sessions and a literature review to identify potential risks.

### Risk matrix and risk assessment

A two-dimensional risk matrix was employed to classify and evaluate potential risks associated with the clinical use of POCT blood glucose meters. Each risk was evaluated according to its probability of harm and severity of harm, following expert consensus and ISO 15189:2022 standards. The matrix (Table 1), following EP23-A standards, provides a systematic approach for prioritizing risks, serving as an initial tool for assessing and prioritizing identified risks (see S1 File for details) [23]. Risks categorized as "unacceptable" necessitated targeted control measures.

Table 1. Risk acceptability matrix.

| Probability of Harm (Definition) | Severity of Harm | | | | |
|---|---|---|---|---|---|
| | Negligible[a] | Minor[b] | Serious[c] | Critical[d] | Catastrophic[e] |
| **Frequent** (once per week.) | Unacceptable | Unacceptable | Unacceptable | Unacceptable | Unacceptable |
| **Probable** (once per month.) | Acceptable | Unacceptable | Unacceptable | Unacceptable | Unacceptable |
| **Occasional** (once per quarter) | Acceptable | Acceptable | Acceptable | Unacceptable | Unacceptable |
| **Remote** (once per year.) | Acceptable | Acceptable | Acceptable | Unacceptable | Unacceptable |
| **Improbable** (once every 5 years or more.) | Acceptable | Acceptable | Acceptable | Acceptable | Acceptable |

[a]Negligible: No impact on patient safety or clinical decisions.

[b]Minor: Minimal impact requiring repeat testing or minor delays, no harm to patient.

[c]Serious: Moderate impact causing diagnostic delays or inaccuracies, not life-threatening.

[d]Critical: Significant impact leading to diagnostic errors or inappropriate treatment, potential patient harm.

[e]Catastrophic: Severe impact, such as missed critical condition diagnosis, resulting in permanent harm or death.

### Risk control and effectiveness evaluation

Control measures were implemented for the identified unacceptable-risk factors based on their risk levels. A follow-up risk assessment conducted three months later evaluated the effectiveness of these measures in both hospitals. Control measures included feedback collection, quality control data analysis, and incident report reviews. This iterative process ensured the practicality and effectiveness of implemented controls in mitigating the risks associated with POCT blood glucose meters. In the effectiveness assessment, inadequate refers to conditions where the standard is entirely unmet, while insufficient refers to conditions where partial compliance exists but improvements are necessary.

## Result

### Risk assessment of the POCT blood glucose monitoring process

The risk assessment revealed that Peking University Shenzhen Hospital faces major risks in three areas: inadequate performance verification prior to hospital entry, insufficient personnel training, and data management issues. These risks can lead to inconsistencies and errors in the monitoring process, ultimately affecting result accuracy. In contrast, Wuhan Third Hospital primarily encounters risks related to insufficient personnel training, directly impacting operational safety and accuracy. Furthermore, Wuhan Hospital faces heightened risks associated with insufficient calibration and inadequate quality control. Identifying these distinct risk factors provides targeted information for subsequent control measures (see Table 2).

### Control measures for unacceptable risks and re-evaluation

To address the identified unacceptable factors, targeted control measures were implemented. At Peking University Shenzhen Hospital, measures focused on inadequate equipment performance verification involved stringent validation protocols to ensure thorough assessment of all equipment prior to use. Additionally, to combat insufficient personnel training, a comprehensive training system and assessment mechanism were established to ensure all operators are adequately qualified. In data management, the hospital introduced automated IT systems

**Table 2. Risk assessment of the POCT blood glucose monitoring process.**

| Aspect | Identified risks | Risk control measures | Risk Level (Peking University Shenzhen Hospital) (Probability, Severity) | Risk Level (Wuhan Third Hospital) (Probability, Severity) |
|---|---|---|---|---|
| **Pre-Analytical** | | | | |
| Management | Unclear management framework and undefined responsibilities. | 1. Establish an internal POCT committee to delineate responsibilities and management processes.<br>2. Conduct regular meetings to review and update POCT management policies. | Acceptable (Remote, Serious) | Acceptable (Remote, Serious) |
| Equipment Evaluation and Selection | Inadequate performance verification prior to hospital entry. | 1. Implement stringent performance verification during equipment introduction, including pre-purchase performance Validation, focusing on critical parameters such as hematocrit and measurement range [W].<br>2. Establish performance verification procedures [W].<br>3. Maintain detailed records with periodic audits [W]. | Unacceptable (Occasional, Critical) | Acceptable (Improbable, Critical) |
| Multiple Instrument Brands | Challenges arising from the use of various brands leading to inconsistencies. | 1. Develop standardized operating procedures for each brand.<br>2. Cross-train staff on different instruments to ensure operational consistency. | Acceptable (Occasional, Serious) | Acceptable (Occasional, Serious) |
| Environmental | Non-compliance with environmental conditions. | Install environmental monitoring systems. | Acceptable (Remote, Serious) | Acceptable (Remote, Serious) |
| **Analytical** | | | | |
| Document and records of SOP | Lack of standardized SOPs and absence of appropriate regulations for POCT management. | 1. Develop standardized operating procedures (SOPs).<br>2. Regularly review and update SOPs.<br>3. Maintain complete records of POCT processes or activities. | Acceptable (Occasional, Minor) | Acceptable (Occasional, Minor) |
| Personnel Training | Insufficient personnel training | 1. Develop a comprehensive training system.<br>2. Implement a tiered qualification system.<br>3. Strengthen oversight of interns and trainees. | Unacceptable (Probable, Serious) | Unacceptable (Probable, Serious) |
| Patient Identification Verification | Patient misidentification. | Establish procedures for verifying patient identity;<br>Operators should verify patient identity before starting POCT. | Acceptable (Improbable, Critical) | Acceptable (Improbable, Critical) |
| Improper Specimen Handling | Inadequate protocols for specimen management. | 1. Develop specimen handling protocols.<br>2. Train staff on proper techniques.<br>3. Monitor adherence to handling protocols. | Acceptable (Occasional, Serious) | Acceptable (Occasional, Serious) |
| Instrument | Equipment failure or sensor malfunction. | Conduct regular inspections and maintenance. | Acceptable (Remote, Serious) | Acceptable (Remote, Serious) |
| Reagents | Reagent failure or batch variation in reagents, | Strictly monitor storage conditions and establish an expiration tracking system. Validate performance of new reagent batches for consistency. | Acceptable (Remote, Serious) | Acceptable (Remote, Serious) |
| Calibration | Insufficient calibration | 1. Establish a clear calibration plan and conduct additional calibration checks as needed [P]. | Acceptable (Occasional, Serious) | Unacceptable (Probable, Serious) |
| Internal Quality Control (IQC) | Inadequate quality control; daily QC not performed as required; delayed response to failures. | Establish IQC concentration levels and QC frequency before using new equipment.<br>2. Review IQC results before each test.<br>3. Document standards for IQC acceptance and rejection.<br>4. Establish quarterly oversight by the POCT committee [P]. | Acceptable (Occasional, Serious) | Unacceptable (Probable, Serious) |
| External Quality Assessment (EQA) | Failure to implement external quality assessment or comparison programs. | 1. Implement external quality assessment<br>2. Seek alternative methods for comparison with results from biochemical analyzers in medical laboratories. | Acceptable (Remote, Serious) | Acceptable (Remote, Serious) |
| Data Management | Manual errors, patient misidentification, or data loss. | 1. Patient information and test results are entered manually and verified. | Unacceptable (Probable, Serious) | Unacceptable (Probable, Serious) |
| QIs | Insufficient ongoing review of quality indicators. | Implement routine review and monitoring of QIs according to ISO 15189:2022 guidelines. | Acceptable (Occasional, Serious) | Acceptable (Occasional, Serious) |

*(Continued)*

**Table 2.** (Continued)

| Aspect | Identified risks | Risk control measures | Risk Level (Peking University Shenzhen Hospital) (Probability, Severity) | Risk Level (Wuhan Third Hospital) (Probability, Severity) |
|---|---|---|---|---|
| **Post-Analytical** | | | | |
| Result Reception | Test results not reported to appropriate personnel. | 1. Ensure that test results are reported to the requester or the patient; 2. Follow standardized operating procedures for reporting results. | Acceptable (Remote, Minor) | Acceptable (Remote, Minor) |
| Result Interpretation | Inappropriate interpretation of results. | 1. Interpret test results according to manufacturer instructions 2. Clinicians should provide clinical consultation and referral recommendations. | Acceptable (Remote, Serious) | Acceptable (Remote, Serious) |
| Interferences | Potential interferences affecting glucose measurement accuracy. | 1. Develop guidelines for recognizing common interferences (e.g., hematocrit levels, medications). 2. Conduct operator training to enhance awareness and understanding of interferences 3. Consult laboratory professionals for advice on interpreting results. | Acceptable (Remote, Serious) | Acceptable (Remote, Serious) |
| Result Reporting | Incomplete elements in POCT reports, lacking essential information. | 1. Ensure report design complies with ISO 15189:2022, including essential elements such as: - Patient information - Operator information - Detection time - Test results - Test type as POCT. 2. Regularly review report formats to ensure completeness and compliance with standards [P]. | Acceptable (Occasional, Serious) | Acceptable (Occasional, Serious) |
| Critical Value Reporting | Delayed or incorrect reporting of critical values. | 1. Immediately report critical values to the physician and ensure timely action; 2. Collect venous blood for retesting in the laboratory if critical values are detected. | Acceptable (Improbable, Catastrophic) | Acceptable (Improbable, Catastrophic) |
| Biosafety and Infection Control | Failure to implement reasonable protective measures. | 1. Personnel involved in sample collection and handling should be equipped with personal protective equipment; hand hygiene should be practiced at least before and after patient contact, between contacts with different patients, and after glove removal; 2. Use manufacturer-recommended cleaning agents for POCT equipment; safely handle and dispose of all samples, collection devices (e.g., needles, reagents, and kits.) | Acceptable (Occasional, Serious) | Acceptable (Occasional, Serious) |

[P]Peking University Shenzhen Hospital has additional risk control measures compared to Wuhan Third Hospital.

[W]Wuhan Third Hospital has additional risk control measures compared to Peking University Shenzhen Hospital.

to minimize human errors and enhance data entry accuracy. Conversely, Wuhan Third Hospital emphasized reinforcing personnel training and oversight, ensuring new employees complete thorough training and assessments to mitigate risks stemming from inadequate training. For quality control, Wuhan Hospital strengthened daily quality control checks and established rigorous calibration plans, adjusting risk levels to acceptable. The implementation of these control measures and subsequent evaluations aims to reduce unacceptable-risk levels and ensure the safety and reliability of the POCT blood glucose monitoring process (see Table 3).

Following these improvement measures, ongoing monitoring of the POCT blood glucose monitoring process has ensured that all associated risks remain within acceptable levels. This proactive approach enhances the overall reliability and safety of the monitoring process, thereby improving patient care outcomes.

## Discussion

This study systematically employed Failure Mode and Effects Analysis (FMEA) to assess the clinical use of point-of-care testing (POCT) blood glucose meters at two hospitals, aligning with ISO 15189:2022. The analysis identified critical risks, including inadequate performance

**Table 3. Control measures and re-evaluation of unacceptable risks.**

| Unacceptable-Risk Identified | Peking University Shenzhen Hospital | Wuhan Third Hospital |
|---|---|---|
| 1. Inadequate performance verification prior to hospital entry. | Post-control Risks<br>Acceptable (Improbable, Critical)<br>Control Measures<br>1. Establish Rigorous Performance Verification Protocols: Implement standardized protocols for performance verification of new equipment before clinical use, focusing on critical parameters.<br>2. Conduct Pre-Purchase Performance Assessments: Assess equipment performance in controlled settings prior to purchase, ensuring it meets clinical needs.<br>3. Regular Equipment Cross-Verification: Periodic cross-checks with standard laboratory analyzers to ensure consistency and reliability.<br>4. Implement Real-time Monitoring System: Utilize IT systems for real-time monitoring to detect and address performance issues promptly.<br>5. Ongoing Training and Re-validation: Regular training for staff on equipment operation and periodic re-validation to ensure competency. | N/A |
| 2. Insufficient Personnel Training | Post-control Risks<br>Acceptable (Remote, Serious)<br>Control Measures<br>1. IT-driven Personnel Management: Automate tracking of personnel certification and training status, with automated reminders for re-certification and refresher courses.<br>2. Comprehensive Onboarding Program: Ensure new employees undergo both theoretical and operational training, covering all relevant POCT procedures.<br>3. Evaluation and Certification: Implement assessments of new staff, with certification awarded upon successful completion of both theoretical and practical exams.<br>4. Pre-Operational Assessment and Authorization: Require pre-operational assessments and ensure only authorized, trained personnel can access the system, using mandatory system login to verify qualifications. | Post-control Risks<br>Acceptable (Occasional, Serious)<br>Control Measures<br>1. Manual tracking of personnel: Certification and training status monitored via logbooks, requiring periodic audits.<br>2. Enhanced Supervision with Mentorship: Establish a structured mentorship program where experienced staff provide hands-on training and continuous oversight to new personnel, ensuring proper skill development and adherence to protocols.<br>3. Ongoing Performance Assessments: Conduct regular, scheduled assessments of operational competence to ensure continuous improvement and compliance with training standards. |
| 3. Insufficient Calibration | N/A | Post-control Risks<br>Acceptable<br>(Occasional, Serious)<br>1. Establish Routine Calibration Schedule: Annual calibration with documentation.<br>2. Follow Manufacturer Guidelines: Ensure calibration follows recommended intervals<br>3.Performance Monitoring: Trend analysis for calibration accuracy and adjustments. |
| 4. Inadequate Quality Control (QC) | N/A | Post-control Risks<br>Acceptable (Occasional, Serious)<br>Control Measures<br>Regular training and competency assessments: Enhance training sessions for staff on QC protocols.<br>Establish quarterly oversight by the POCT committee. |
| 5. Data Management | Post-control Risks<br>Acceptable (Improbable, Serious)<br>Control Measures<br>1. Automated data management: IT systems automatically record test results and patient information.<br>2. Barcode/RFID patient identification: Automatically link patient data to records, ensuring accurate data collection.<br>3. Real-time data backup: IT-based backup ensures secure and immediate data recovery. | Post-control Risks<br>Acceptable (Remote, Serious)<br>Control Measures<br>1. Manual verification: Patient identity and data must be manually checked and verified by staff.<br>2. Regular manual audits: Staff conduct audits of records to ensure accuracy.<br>3. Paper-based backups: Data recorded on paper and manually filed. |

verification prior to hospital entry, insufficient personnel training, data management deficiencies, insufficient calibration and inadequate quality control (QC) [10,11]. Based on the evaluations provided by the multidisciplinary team during the follow-up period, the implementation

of targeted control measures not only mitigates these risks but also enhances the overall reliability of POCT monitoring, underscoring the necessity of a structured risk management strategy.

Inadequate performance verification at Peking University Shenzhen Hospital highlighted the need for stringent protocols to ensure thorough assessment of all equipment before clinical use. Comprehensive performance verification protocols were established to ensure compliance with necessary standards, aiming to minimize risks of inaccurate readings and enhance patient safety.

Data management issues posed significant risks at Peking University Shenzhen Hospital due to its higher workload and previous reliance on manual processes. To enhance data integrity, integrating information technology (IT) was a critical improvement measure. Automated data entry systems and QC tracking were implemented, significantly reducing the potential for human error associated with manual data entry, which can compromise patient data integrity [24]. Due to limitations in resources and infrastructure availability at Wuhan Third Hospital, manual verification remains the current practice. To mitigate actual risks in resource-limited environments like Wuhan Third Hospital, the risk management team recommends maintaining manual verification, regular manual audits, and paper-based backups. These measures aim to ensure the accuracy and integrity of data management processes in the absence of fully implemented IT systems.

Personnel training emerged as a pivotal risk factor at both hospitals. Insufficient training can lead to improper use of POCT devices, resulting in unreliable test results. To address this, comprehensive training programs and ongoing education initiatives were instituted, equipping clinical staff to operate POCT devices correctly and adhere to standardized operating procedures [2]. These efforts emphasize the importance of continuous professional development in maintaining high-quality testing standards. As with the measures in data management, IT-driven solutions in personnel training can enhance the efficiency and accuracy of training management, ultimately improving the quality of POCT services. However, in resource-limited environments, the following measures remain essential to mitigate risks: manual tracking of personnel, enhanced supervision with mentorship, and ongoing performance assessments.

At Wuhan Third Hospital, inadequate calibration and quality control compounded risks. To address this, a comprehensive calibration schedule was established, incorporating daily internal quality control (IQC) checks, and performance verification tests [8]. A digital log system was implemented to track calibration activities, with alerts for upcoming requirements. This structured approach not only ensures compliance with ISO 15189:2022 standards but also reinforces the reliability of POCT results by enabling cross-verification with standard laboratory analyzers. Additionally, routine comparisons with biochemical analyzers and robust QC measures, including automated data logging and comprehensive internal and external QC protocols, effectively mitigated risks [25]. Post-control evaluations confirmed that insufficient QC measures were reduced to acceptable levels, demonstrating the value of ongoing quality control and multidisciplinary collaboration in POCT risk management.

Effective risk management requires interdisciplinary cooperation. Laboratory professionals are crucial in developing quality control protocols, calibrating instruments, and selecting appropriate POCT devices [22]. Their expertise in identifying potential interferences ensures the accuracy and reliability of POCT results [26]. Nurses and physicians rely on laboratory professionals to investigate discrepancies between POCT and central laboratory results, involving cross-checking device performance and assessing factors that may influence test accuracy.

However, the current study faces several limitations. Firstly, the Failure Mode and Effects Analysis (FMEA) method used in this study relies on clinical expertise and consensus-based brainstorming by a multidisciplinary team. This approach focuses on qualitative risk identification rather than extensive quantitative data collection, which is a recognized limitation of FMEA. Secondly, the data used in this study were obtained from a clinical laboratory setting, which simulates a real-world clinical environment but does not fully capture the variability and complexities of actual clinical practice. While the control measures implemented were effective, future studies should focus on performance indicators such as error rates, accuracy, and patient outcomes to further validate these measures and inform best practices for POCT use. Moreover, future research should explore emerging risks related to the integration of POCT devices into electronic health records (EHRs) and hospital information systems, including challenges related to data security and system interoperability [21]. Leveraging big data analytics and artificial intelligence (AI) could enable real-time monitoring and proactive risk detection [27], ultimately improving the reliability of POCT.

## Conclusions

This study highlights the effectiveness of Failure Mode and Effects Analysis (FMEA) in identifying and mitigating risks associated with point-of-care testing (POCT) blood glucose meters. Key improvements, including structured performance verification, comprehensive training, and enhanced data management through information technology, have noticeably improved accuracy and reliability. Interdisciplinary collaboration among laboratory professionals and clinical staff is essential for ensuring proper use and data integrity. Future research should focus on emerging risks with electronic health records and leverage big data analytics to further enhance POCT safety and patient care outcomes.

## Supporting information

**S1 File. Risk acceptability matrix discussion.**
(PDF)

**S2 File. Complete risk assessment report: Pre-analytical, analytical, and post-analytical phases.**
(PDF)

## Acknowledgments

We extend our gratitude to the medical staff and laboratory professionals at Peking University Shenzhen Hospital and Wuhan Third Hospital for their invaluable contributions and collaboration in this study. Their expertise was crucial in implementing the Failure Mode and Effects Analysis (FMEA) and enhancing the safety of point-of-care testing (POCT) blood glucose meters. We also thank our research team for their support, as well as our colleagues for their insightful feedback. Their expertise and cooperation were crucial to the risk assessment and quality control processes.

## Author contributions

**Conceptualization:** Xiagang Luan, Lingling Ke, Hao Xue, Yong Xia.

**Data curation:** Xiagang Luan, Lingling Ke.

**Investigation:** Xiagang Luan, Lingling Ke, Minxuan Feng.

**Project administration:** Weiqun Peng, Houlong Luo.

**Writing – original draft:** Xiagang Luan, Lingling Ke, Minxuan Feng.

**Writing – review & editing:** Hao Xue, Yong Xia.

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
