## [Decision Letter · Decision Letter 0]

6 Jan 2025

PONE-D-24-51452Risk Management in POCT Blood Glucose Monitoring: FMEA Approach Aligned with ISO 15189:2022PLOS ONE

Dear Dr. Xia,

Thank you for submitting your manuscript to PLOS ONE. After careful consideration, we feel that it has merit but does not fully meet PLOS ONE’s publication criteria as it currently stands. Therefore, we invite you to submit a revised version of the manuscript that addresses the points raised during the review process.

We look forward to receiving your revised manuscript.

Kind regards,

Vittorio Sambri, M.D., Ph.D.

Academic Editor

PLOS ONE

Journal Requirements:

2. We note that your Data Availability Statement is currently as follows: 

“All relevant data are within the manuscript and its Supporting Information files.”

**Additional Editor Comments:**

In detail, your manuscript has been evaluated by two experts in the field of the use of POCT and bio

th found your study interesting. I fully agree with their suggestions and I encourage you to take into consideration their suggestion in oprder to improve the quality and value of your paper.

Reviewers' comments:

Reviewer's Responses to Questions

**Comments to the Author**

1. Is the manuscript technically sound, and do the data support the conclusions?

Reviewer #1: Yes

Reviewer #2: No

2. Has the statistical analysis been performed appropriately and rigorously? 

Reviewer #1: Yes

Reviewer #2: I Don't Know

3. Have the authors made all data underlying the findings in their manuscript fully available?

Reviewer #1: Yes

Reviewer #2: No

4. Is the manuscript presented in an intelligible fashion and written in standard English?

Reviewer #1: Yes

Reviewer #2: Yes

5. Review Comments to the Author

Reviewer #1: The use of POCT systems in laboratory medicine is increasingly widespread.

Having correct and rapid laboratory data brings a great advantage to the clinician. The manuscript analyzes the critical issues of these systems and provides useful information. It's a good article.

Reviewer #2: General: Well written

Specific:

Line 29: ”..in clinical settings….” – however, the study was not conducted in a clinical setting - but in a clinical laboratory setting (line 111), which can only simulated a “real” clinical setting. So, the study does not reflect “real life” POCT. This should be clarified and discussed. Clearly, a risk assessment conducted by certified laboratory staff I a clinical laboratory setting is different from a risk assessment in a real clinical setting (e.g., in an ER).

Line 141 / Table1: Probability – the differences should be described – are the different probabilities based on actual frequencies (e.g., 10-6, etc) or purely descriptive? Severity: The description of the five different severity levels should be included – e.g., Catastrophic: Is the definition of this severity “Death or life-threatening injury or illness” – or?

Line 154-159: “Inadequate” and “insufficient” appears several places. What is the difference?

Line 161 / Table 2: Are the described probabilities based on actual observations / data – or estimates?

Line 161 / Table 2: You mention “validation” on several occasions. Do you mean “verification”?

Line 161 / Table 2 / Analytical-Data Management: The implemented risk control measure is “Patient information and test results are entered manually and verified”. Why not use scanning of barcodes and integration with LIS? This will reduce / remove human errors?

Line 181 / Table 3 / 2. Insufficient personnel training: The implemented risk control measure is “IT driven” for Peking and “Manual tracking” for Wuhan. Why different? Why not implement IT driven in Wuhan also?

Line 181 / Table 3 / 5. Data management: The implemented risk control measure is “automated” for Peking and “Manual verification” for Wuhan. Why different? Why not implement automated in Wuhan also?

Line 193: “…but also enhanced the overall reliability of POCT….”: How do you support this claim? Data? Subjective evaluation?

Line 200-206: If manual data entry was a risk at Peking (and had to be mitigated), why was it not considered a risk at Wuhan?

Line 243-251: You sate “….highlights the efficiency of FMEA…..” and “….have significantly improved….”. Where is the data to support these claims? “Significantly” is a bold claims that requires data

6. PLOS authors have the option to publish the peer review history of their article (what does this mean? ). If published, this will include your full peer review and any attached files.

**Do you want your identity to be public for this peer review?** For information about this choice, including consent withdrawal, please see our Privacy Policy .

Reviewer #1: **Yes: ** Torello Monica

Reviewer #2: No

---

## [Author Response · Author response to Decision Letter 0]

22 Jan 2025

In response to the reviewers’ feedback:

Methodological clarity: We have revised the description of the study setting to clarify that the research was conducted in a simulated clinical laboratory setting. Additionally, we included a discussion on the limitations of using such a setting compared to real-life clinical environments, such as emergency rooms.

Risk assessment details: The descriptions of probability levels and severity classifications in Table 1 have been expanded. Clear definitions for each level (e.g., “Catastrophic” meaning death or life-threatening injury) are now provided to enhance the transparency and interpretability of the risk assessment.

Terminology precision: We have clarified the distinction between “validation” and “verification” in accordance with ISO 15189:2022 standards, and have updated the manuscript by replacing the relevant instances of “validation” with “verification,” ensuring consistency and accuracy throughout the manuscript.

Data management improvements: Due to limitations in resources and infrastructure availability at Wuhan Third Hospital, manual verification remains the current practice. The risk management team have proposed a gradual implementation of automated data management systems in resource-limited settings at Wuhan Third Hospital.

Subjective evaluations: Statements regarding improvements (e.g., "significantly improved" has been replaced with “noticeably improved” in the results section) have been revised to reflect that they were based on expert team feedback. We have emphasized the need for future studies to validate these findings with quantitative data.

FMEA limitations: We have elaborated on the limitations of the FMEA method, including its reliance on expert consensus and the lack of detailed quantitative data. This discussion has been added to the "Discussion" sections.

---

## [Editor Report · Decision Letter 1]

10 Feb 2025

Risk Management in POCT Blood Glucose Monitoring: FMEA Approach Aligned with ISO 15189:2022

PONE-D-24-51452R1

Dear Dr. Xia,

We’re pleased to inform you that your manuscript has been judged scientifically suitable for publication and will be formally accepted for publication once it meets all outstanding technical requirements.

Kind regards,

Vittorio Sambri, M.D., Ph.D.

Academic Editor

PLOS ONE

---

## [Editor Report · Acceptance letter]

PONE-D-24-51452R1

PLOS ONE

Dear Dr. Xia,

I'm pleased to inform you that your manuscript has been deemed suitable for publication in PLOS ONE. Congratulations! Your manuscript is now being handed over to our production team.

Kind regards,

on behalf of

Professor Vittorio Sambri

Academic Editor

PLOS ONE